# The Evolution of Effort-Reward Imbalance in Workers during the COVID-19 Pandemic in France—An Observational Study in More than 8000 Workers

**DOI:** 10.3390/ijerph19159113

**Published:** 2022-07-26

**Authors:** Louis Delamarre, Salma Tannous, Ines Lakbar, Sébastien Couarraze, Bruno Pereira, Marc Leone, Fouad Marhar, Julien S. Baker, Reza Bagheri, Mickael Berton, Hana Rabbouch, Marek Zak, Tomasz Sikorski, Magdalena Wasik, Hijrah Nasir, Binh Quach, Jiao Jiao, Raimundo Aviles, Maëlys Clinchamps, Fréderic Dutheil

**Affiliations:** 1LaPSCo, Physiological and Psychosocial Stress, Université Clermont Auvergne, CNRS, 63000 Clermont-Ferrand, France; salma.tannous@uca.fr (S.T.); fouad.marhar@gmail.com (F.M.); mickael.berthon@uca.fr (M.B.); mclinchamps@chu-clermontferrand.fr (M.C.); frederic.dutheil@uca.fr (F.D.); 2Department of Anesthesiology and Intensive Care, University Hospital of Marseille, Hopital Nord, Assistance Publique Hôpitaux de Marseille, 13015 Marseille, France; ines.lakbar@ap-hm.fr (I.L.); marc.leone@ap-hm.fr (M.L.); 3Pôle Régional d’Enseignement et de Formation aux Métiers de la Santé, University Hospital of Toulouse, 31000 Toulouse, France; couarraze.sebastien@wanadoo.fr; 4Direction de la Recherche Clinique et de l’Innovation, Centre Hospitalier Universitaire de Clermont-Ferrand, 63000 Clermont-Ferrand, France; bpereira@chu-clermontferrand.fr; 5Centre for Health and Exercise Science Research, Department of Sport, Physical Education and Health, Hong Kong Baptist University, Kowloon Tong, Hong Kong, China; jsbaker@hkbu.edu.hk; 6Department of Exercise Physiology, University of Isfahan, Isfahan 81746-73441, Iran; will.fivb@yahoo.com; 7Institut Supérieur de Gestion de Tunis, Université de Tunis, Tunis 2000, Tunisia; hana.rabbouch@gmail.com; 8The Institute of Health Sciences, Collegium Medicum, Jan Kochanowski University of Kielce, ul. Zeromskiego 5, 25-369 Kielce, Poland; mkzak@ujk.edu.pl; 9Doctoral School, Collegium Medicum, Jan Kochanowski University of Kielce, Zeromskiego 5, 25-369 Kielce, Poland; tomasz.sikorski@phd.ujk.edu.pl (T.S.); magdawasik95@gmail.com (M.W.); 10Department of Economic Development, Université Clermont Auvergne, 63000 Clermont-Ferrand, France; hijrahnasir2013@gmail.com; 11Centre for Health and Exercise Science Research, Hong Kong Baptist University, Hong Kong, China; bquach@hkbu.edu.hk (B.Q.); jojojiao@hkbu.edu.hk (J.J.); 12Universidad Finis-Terrae, El-Carmen, Hospital Dr. Luis-Valentìn-Ferrada, Obstetrics and Gynecology, Maipù 5641235, Chile; raimundo.aviles@gmail.com; 13Department of Preventive and Occupational Medicine, University Hospital of Clermont-Ferrand, 63000 Clermont-Ferrand, France

**Keywords:** SARS-CoV-2, France, work-related stress, Siegrist’s framework, lockdowns

## Abstract

(1) Background: The effects of lockdown repetition on work-related stress, expressed through Effort-Reward Imbalance (ERI), during the COVID-19 pandemic are poorly documented. We investigated the effect of repetitive lockdowns on the ERI in French workers, its difference across occupations, and the change in its influencing factors across time. (2) Methods: Participants were included in a prospective cross-sectional observational study from 30 March 2020 to 28 May 2021. The primary outcome was the ERI score (visual analog scale). The ERI score of the population was examined via Generalized Estimating Equations. For each period, the factors influencing ERI were studied by multivariate linear regression. (3) Results: In 8121 participants, the ERI score decreased in the first 2 lockdowns (53.2 ± 0.3, *p* < 0.001; 50.5 ± 0.7, *p* < 0.001) and after lockdown 2 (54.8 ± 0.8, *p* = 0.004) compared with the pre-pandemic period (59 ± 0.4). ERI was higher in medical than in paramedical professionals in the pre-pandemic and the first 2 lockdowns. Higher workloads were associated with better ERI scores. (4) Conclusions: In a large French sample, Effort-Reward Imbalance worsened during the COVID-19 pandemic until the end of the 2nd lockdown. Paramedical professionals experienced a higher burden of stress compared with medical professionals.

## 1. Introduction

Even outside the healthcare sector, the COVID-19 pandemic has had a huge impact worldwide, due to the shutting of most countries’ economies, the forced change of working routines, and the uncertainties in employment [1,2,3,4,5,6,7,8,9,10,11,12,13,14,15,16]. The constant stream of fearsome news reports may also have exacerbated the anxiety in the population [17,18,19]. As a result, the COVID-19 pandemic may have had considerable effects on stress. Effort-Reward Imbalance (ERI) is one of the models representing work-related stress. As theorized by Siegrist in 1996 [20], social reciprocity describes the fact that cooperation at work leads to the expectancy of a return on the time and effort invested in cooperation with other workers. As such, the perceived stress at work is a function of the subjective balance between the efforts and the rewards perceived by a worker [20]. Siegrist’s model combines two components, an extrinsic one, summarizing the imbalance between efforts and rewards, and an intrinsic one that refers to the perceived over-commitment to work. This second component is thought to be a personality trait rather than a feature related to an occupation or a profession [21]. The association of ERI and work-related stress has already been described [21,22,23,24,25], and is associated with somatic complications, including cardiovascular mortality [24].

Although the effect of lockdown on stress during the COVID-19 pandemic is becoming progressively clearer [2,6,8,26,27], there is scarce data regarding the effects of the repetition of lockdowns on work-related stress and especially ERI. While the general population may have experienced an increase in stress, certain professions may present a particular risk, such as health professionals [4,5,7,9,11,12,27,28]. In the healthcare sector, paramedical professionals are reported to be more at risk of work-related stress due to the COVID-19 pandemic [29,30], but the effect of repetitive lockdowns on this pattern is not known. The factors associated with ERI are poorly documented and the evolution of their relative influence during the pandemic is not described, especially family-related factors such as the number of children [31,32].

We hypothesized that ERI may have worsened across the lockdowns during the COVID-19 pandemic. Therefore, the primary objective of the present study was to investigate the effect of the repetition of lockdowns on the ERI in French workers and its difference between occupations. The secondary objectives of the study were to investigate the factors associated with ERI and the change of their relative influence during the pandemic in France.

## 2. Materials and Methods

### 2.1. Study Design

The COVISTRESS project is an international prospective cross-sectional observational study on the general population during the COVID-19 pandemic. The inclusions began on 30 March 2020 and are still ongoing. We used an anonymous questionnaire, available in ten languages, distributed electronically. The study was approved by the Southeast VI Ethical Committee of France (Clinicaltrials.gov. NCT04538586).

### 2.2. Participants

The international COVISTRESS network distributed the questionnaire online (http://covistress.org/index-en.html, accessed 1 June 2022), without distinction of country, gender, or occupation. The participants were included from 30 March 2020 to 28 May 2021.

### 2.3. Instrument Survey

The main outcome of the present study was Effort-Reward Imbalance score (hereafter designated as ERI score), measured through a visual analog scale, i.e., a non-calibrated horizontal line ranging from minimum (0) to maximum (100) [33]. The survey question was formulated as: “What is your level of satisfaction at work, considering the efforts you put in?”, so the higher the score, the higher the perceived reward compared with the effort produced. 

The time stamp of participants’ response was used to segment the population into subgroups based on the lockdown periods in France, i.e., pre-pandemic period (before 17 March 2020): 

Lockdown 1 (from 17 March 2020 to 11 May 2020); Post-lockdown 1 (from 11 May 2020 to 30 October 2020); Lockdown 2 (from 30 October 2020 to 15 December 2020); Post-lockdown 2 (from 15 December 2020 to 3 April 2021); Lockdown 3 (from 3 April 2021 to 3 May 2021); Post-lockdown 3 (from 3 May 2021 to 28 May 2021). We used those periods to model the ERI score during the pandemic. The occupation of the participants was recorded from a drop-down list of categories (‘Executive and intellectual occupation’, ‘Intermediary profession’, ‘Farmer’, ‘Artisan, merchant, or entrepreneur’, ‘Manual worker’, ‘Student’, ‘Unemployed’, ‘Retired’). They could then choose among a detailed list their exact occupation. Medical professionals were listed in the ’Executive and intellectual occupation’ and paramedical professionals in the ’Intermediary profession’ categories. The results were then aggregated into three classes, i.e., medical professions, paramedical professions, and other occupations. 

The other explored variables were sociodemographic characteristics (age, sex, marital status, number of children). Sex was either Female or Male. Marital status was defined as ‘Single’, ‘Couple’, or ‘Other’. Sex, occupation, and marital status were collected as categorical variables. The number of children, age, and ERI score were collected as quantitative variables. Weekly working hours were collected as a categorical ordinal variable. Those variables were considered as potential explicating factors for the ERI score.

The survey first collected the reported ERI score, and the number of hours worked per week at the time of the respondent’s participation. In a subsequent iteration of the survey, the participants were also asked to declare their number of hours worked per week and their ERI score before the pandemic. Given the anonymous nature of the present survey, the participants’ identification, email address, or contact information was not recorded. As a result, the study population is a collection of prospectively included cross-sectional cohorts. 

### 2.4. Statistical Analysis

No calculation of the sample size was performed *a priori* given the prospective, observational, cross-sectional nature of the present project. The invitation to participate was distributed by the investigators, and the academic and institutional partners at their discretion.

Data was expressed in number and percentage for categorical variables and in mean and standard error (SE) for quantitative variables. Comparisons between categorical variables were made using Chi2 test (χ^2^). Comparisons of quantitative variables between groups were completed using a Mann–Whitney U test (if two categories were considered); or a Kruskal–Wallis test (in case of more than two categories) followed by Dunn’s test for pairwise comparisons in case of significant differences, with *p*-value adjustment based on the Bonferroni method. The correlation between continuous variables was assessed through the Pearson correlation coefficient. Observations with missing data were omitted for statistical analysis, without any imputation. 

To examine the effect of time (the lockdown periods) and other variables (age, sex…) on the reported ERI score at the population level, we used Generalized Estimating Equations (GEE) [34,35,36]. For GEE, time was expressed as a numerical value. Various correlation structures were tested (independence, exchangeable, unstructured, and autoregressive). The models yielded by those different correlation structures were compared through QIC(I) to select the best model [37]. To further examine the validity of the GEE models, we performed a sensitivity analysis on a subpopulation constituted of the participants where repeated measurements of ERI score was available for the pre-pandemic, lockdown 1 and lockdown 2 periods. The second-order interactions between independent variables were tested and reported.

Finally, for each of those main periods (pre-pandemic period, lockdown 1, and lockdown 2), the factors associated with the ERI score were investigated by bivariate regressions followed by multivariate linear regression. A variable was included in the multivariate linear regression model if its *p*-value was <0.1 in bivariate regression. The final model was obtained using a mixed selection. Model performance was measured by the Residual Square Error and the adjusted R^2^.

Quantitative variables were grouped into classes when necessary. Considering the number of children declared by the participants, we created classes of ’0’, ’1’, ’2’, ’3’, and ’4 or more’ children, based on the epidemiological data published by the French National Institute of Statistics and Economic Studies [38]. The discretization of the number of hours worked per week into broader classes of ‘<30 h’, ‘30–40 h’, ‘40–50 h’, and ‘>50 h’ was based on the analysis of the INSEE open database where the classes ‘30–40 h’ and ‘40–50 h’ accounted for most of the citizens [39]. The age variable was discretized into classes of ‘<35 years’, ‘35–45 years’, ‘45–55 years’, ‘55–65 years’, and ‘>65 years’.

Except for the occupation, no subgroup analyses were performed based on the participants’ profile. The statistical analysis plan was decided before analysis. A value of *p* < 0.05 was required for statistical significance and statistical tests were two-tailed. Analyses were performed using R software 4.0.4 (R Core Team, R Foundation for Statistical Computing, Vienna, Austria, 2021) [40]. The STROBE checklist for cross-sectional studies was used to report the results of the present study [41]. 

## 3. Results

The French COVISTRESS cohort had 12,079 participants, of whom 11,874 were eligible for analysis. Those participants’ characteristics are presented in Table 1. The flowchart of the study is presented in Figure 1.

### 3.1. Variations of ERI during the Pandemic

The ERI score differed significantly across the different periods considered (*p* < 0.001). The ERI score decreased in the following periods compared with the pre-pandemic period (59 ± 0.4): lockdown 1 (53.2 ± 0.3, *p* < 0.01), lockdown 2 (50.5 ± 0.7, *p* < 0.01), post-lockdown 2 (54.8 ± 0.8, *p* < 0.01).

The ERI score was also lower during lockdown 2 compared with lockdown 1 (50.5 ± 0.7 versus 53.2 ± 0.3, respectively, *p* = 0.01) and higher during post-lockdown 2 compared with lockdown 2 (54.8 ± 0.8 versus 50.5 ± 0.7, respectively, *p* < 0.01). These results are presented in Figure 2.

### 3.2. Variations of ERI by Occupation

In medical professionals (Figure 3A), ERI differed significantly between periods (*p* = 0.01) with a significantly lower ERI score in post-lockdown 2 compared with the pre-pandemic period (*p* = 0.02). In paramedical professionals (Figure 3B), ERI differed between periods (*p* = 0.01) with a lower ERI score during lockdown 2 compared with the pre-pandemic period (*p* < 0.01). Finally, in other occupations (Figure 3C), ERI score also differed between periods (*p* < 0.01), with a lower ERI score during lockdown 1 and lockdown 2 compared with the pre-pandemic period (*p* < 0.01 in both comparisons), and a higher ERI score in the post-lockdown 2 period compared with the lockdown 2 period (*p* = 0.04).

ERI score was higher in medical professionals compared with paramedical professionals and other workers in the pre-pandemic period (62.1 ± 0.6 vs. 56 ± 0.6 and 58.6 ± 0.3, adj. *p* < 0.01 and adj. *p* < 0.01, respectively) and during lockdown 1 (58.4 ± 0.8 vs. 53.3 ± 0.7 and 52 ± 0.3, adj. *p* < 0.01 and adj. *p* < 0.01, respectively) and higher than in paramedical professionals during lockdown 2 (55.1 ± 0.8 vs. 48 ± 0.8, adj. *p* = 0.02).

### 3.3. Factors Influencing ERI during the Pandemic

Generalized Estimating Equations analysis was used to investigate the factors associated with ERI, including time, profession, sex, age, marital status, number of children, and number of hours worked per week as covariates. The GEE model with exchangeable covariance structure (GEE-ex) yielding the best performance, is described in Figure 4. Medical professionals (vs. other occupations, coefficient: 4.28, 95% CI: 2.57 to 6) and high number of hours worked per week (40 to 50 h/week and >50 h/week; 0.7, 0.16 to 1.19 and 1.26, 0.64 to 1.87, respectively) were positively associated with ERI scores. Age between 35 and 45 and between 45 and 55 (−2.19, −4.16 to −0.23 and −2.14, −4.28 to −0.01, respectively, compared with age < 35) and the time spent from the beginning of the pandemic were negatively associated with the ERI score. Significant second-order interactions between the variables considered in the GEE model were found, in particular between sex and number of children, age and number of children. They are presented in Appendix A. A sensitivity analysis was performed on the subgroup in which the ERI score was collected for the pre-pandemic, lockdown 1, and lockdown 2 periods and found an association of the paramedical profession and the time spent from the beginning of the pandemic with worse ERI scores (Appendix A).

### 3.4. Did the influence of factors associated with ERI change during the pandemic?

During the pre-pandemic period, the occupation (*p* < 0.01), number of hours worked per week (*p* < 0.01), age (*p* < 0.01), number of children (*p* < 0.01), and sex (*p* = 0.02) were significantly associated with ERI score in bivariate analysis. In multivariate linear regression, age above 55 years (ages 55–65: 4.9, 2.2 to 7.6; age > 65: 11.1, 5.1 to 17; *p* < 0.01 and *p* < 0.01, respectively) and classes of above-normal number of hours worked per week (40–50 h: 2.89, 0.4 to 5.4; >50 h, 4.36, 1.5 to 7.2; *p* = 0.02 and *p* < 0.01, respectively) were associated with increased ERI scores. The paramedical profession (−2.86, −5.5 to −0.2, *p* = 0.03) was associated with decreased ERI score.

During the first lockdown, the number of children (*p* = 0.049), the occupation (*p* < 0.01), and number of hours worked per week (*p* < 0.01) were associated with the ERI score in bivariate analysis. In multivariate linear regression, age below 35 years (2.2, 0.1 to 4.3, *p* = 0.04), the medical profession (4.1, 2 to 6.3, *p* < 0.01), and number of hours worked per week > 50 h (4.4, 2 to 6.9, *p* < 0.01) were associated with increased ERI scores.

During the second lockdown, only the occupation was associated with the ERI score (*p* = 0.02) in bivariate analysis. In linear regression, the medical profession (vs. other occupations, 4.9, 0.6 to 9.2, *p* = 0.03) was associated with higher ERI scores. Detailed results are presented in Appendix A.

## 4. Discussion

Based on one of the largest samples reported, our results suggest that Effort-Reward Imbalance worsened during the COVID-19 pandemic in France, at least until the end of the second lockdown. Medical professionals seemed relatively protected against this worsening ERI. The ERI was also associated with other factors, the weights of which changed as the pandemic progressed.

### 4.1. How the ERI Has Evolved over the Course of the Pandemic

The ERI changed during the pandemic in France, with a decrease in job satisfaction relative to effort, at least until the end of the second lockdown, with worsened ERI during lockdown 2 compared with lockdown 1, in our population. Our results suggest a worsening of ERI until the second lockdown (15 December 2020), and no respite in terms of work-related stress after the end of the first lockdown. Although other reports have described the rise in stress caused by the pandemic [2,6,8,26,27], no clear data is available elsewhere specifically on the additive effect of lockdowns on work-related stress.

### 4.2. Differences in ERI between Medical and Paramedical Professionals 

Our results also suggest that the medical profession is independently associated with better ERI scores, specifically when compared with paramedical profession in the pre-pandemic period and the first two lockdowns. The higher stress burden on paramedical professionals, compared with medical professionals, has been described during the pandemic [9,12,42]. Even if the pandemic also increases the meaning or purpose in both medical and paramedical professionals [42], evidence suggests that stress is higher for paramedical professionals than for medical professionals [27,29,30,43].

### 4.3. Factors Associated with ERI

Despite evidence pointing to higher stress levels in women compared with men, our results did not show clear differences in Effort-Reward imbalance according to gender [1,4]. Sandoval-Reyes et al. suggested that perceived stress affects men’s productivity more than women’s, suggesting a difference in efforts-rewards perception according to gender [2]. We assume that when other factors are taken into account in a larger sample, these differences may diminish.

Our results suggest that people who continued to work or increased their number of hours worked per week felt somehow more rewarded, given the efforts they provided. This result contrasts with some reports that suggest a connection between high workloads and depression or anxiety [44]. In non-healthcare personnel, the reduction of number of hours worked per week and the extensive use of working from home may have contributed to the observed reduction in the perception of reward vs. efforts. The effects of working from home on perceived stress, work satisfaction, and productivity are variable across studies [2,15]. Due to the near shutdown of many industries, the first and second lockdowns may have led to lower rewards, especially financially, for workers outside the health care sector [10]. Unstructured working time, interruptions due to family or childcare needs during lockdowns, and the technical limitations of working from home may also have contributed to the effort-reward imbalance [3,45].

Even if the participants’ age was not an independent factor associated with ERI in the model we report in the present research, younger age was independently associated with better ERI during the first lockdown, contrasting with other reports suggesting that older adults develop better strategies to cope with stress [6]. During the second lockdown, age was not independently associated with ERI scores. Taking into account the relatively small effect size of this difference, we hypothesize that younger workers, who were less likely to face barriers to career change and health-related constraints regarding SARS-CoV-2, were more likely to maintain their work engagement, at least initially. Finally, contrary to other reports, family-related variables such as marital status or the number of children were not associated with ERI in our population [9,44]. In the existing literature, family responsibilities tends to exert a negative impact on ERI in younger women, while their effect is marginal in older women [31]. Sperlich et al. found a negative association between the number of children and the Effort-Reward Imbalance in women [32]. We did not find a significant effect of the number of children on ERI in our population, but we did not perform a subgroup analysis on women, nor did we take into account the age of the children. It is also possible that the effects of these variables may be attenuated in a large sample and when a large number of explanatory variables are taken into account.

### 4.4. How Time Modifies the Strength of the Association between ERI and Its Associated Factors

Our results suggest that the influence of the factors associated with ERI varied as the pandemic progressed. While age, the type of occupation, and the number of hours worked per week were independently associated with ERI scores in the pre-pandemic period and the first lockdown, only the type of occupation was independently associated with ERI in the second lockdown in France. No similar data is available regarding the evolution of the determinants of ERI during the pandemic, thus limiting the external validity of this result. 

These findings suggest that, even if lockdowns may reduce the spread of a pandemic, their additive effect may be deleterious to the mental health and work-related stress of workers. In addition, paramedical professionals may be at greater risk of Effort-Reward imbalance compared with medical professionals. We hypothesize that medical professionals may experience more meaning at work, even in pandemic-related extreme situations, relative to the efforts they provide. These results bring new light to many alarming signals, reporting an increasing difficulty to hire and retain paramedical professionals in French hospitals [46,47,48].

### 4.5. Limitations

This study has limitations. First, the participants contributed anonymously, and no sampling was done to ensure comparability with the general population, thus leading to a risk of response bias. Similarly, we did not specifically analyze the level of exposure to SARS-CoV-2 nor did we sample the population to explore this feature, which may have been of interest in both the healthcare and non-healthcare sectors [11].

Second, some questions may be at risk of recall bias, because no prior data regarding the work and work-related stress of the participants was collected before the pandemic. Given the anonymous nature of the survey, we did not obtain repeated measures throughout the pandemic from respondents. This trade-off between repeatability and anonymity was in favor of anonymity as it allowed a higher number of responses needed for this analysis. Third, we focused on the French population. The results of this study may thus not be generalizable to other countries, especially those with strong differences in the pandemic dynamics, national lockdown modalities, or governmental compensations for financial loss. Finally, the use of visual scales may lead to a risk of self-reporting bias. Nevertheless, this approach has already been used with various instruments.

This study also has strengths. First, despite the risks of the aforementioned biases, the large number of participants allows for a relatively good generalization to medical and paramedical professionals in the French health care system. We can also suppose that the anonymous nature of the survey enhanced the return rate of questionnaires and may also have allowed for more accurate answers, given the relatively personal or sensitive nature of some of the questions asked. Second, due to the variety of participants, this study was able to compare work-related stress between healthcare professionals and workers in other occupations, which is often lacking in this field.

Finally, our results show that maintaining the number of hours worked per week by workers during the pandemic was associated with a higher perception of Effort-Reward, consistent with other reports. This supports the French policy of almost compulsory working from home and subsidised part-time work in most French industries. The latter allows an employer in difficulty to have all or part of the cost of the remuneration of its employees covered, thus limiting the risk of job termination [49].

## 5. Conclusions

As hypothesized, work-related stress, expressed through the Effort-Reward imbalance framework, was exacerbated during the first two French lockdowns. In the health care sector, paramedical professionals seem to have been more affected than medical professionals. Reduced working time during lockdowns was associated with a worsening of the Effort-Reward imbalance.

## Figures and Tables

**Figure 1 ijerph-19-09113-f001:**
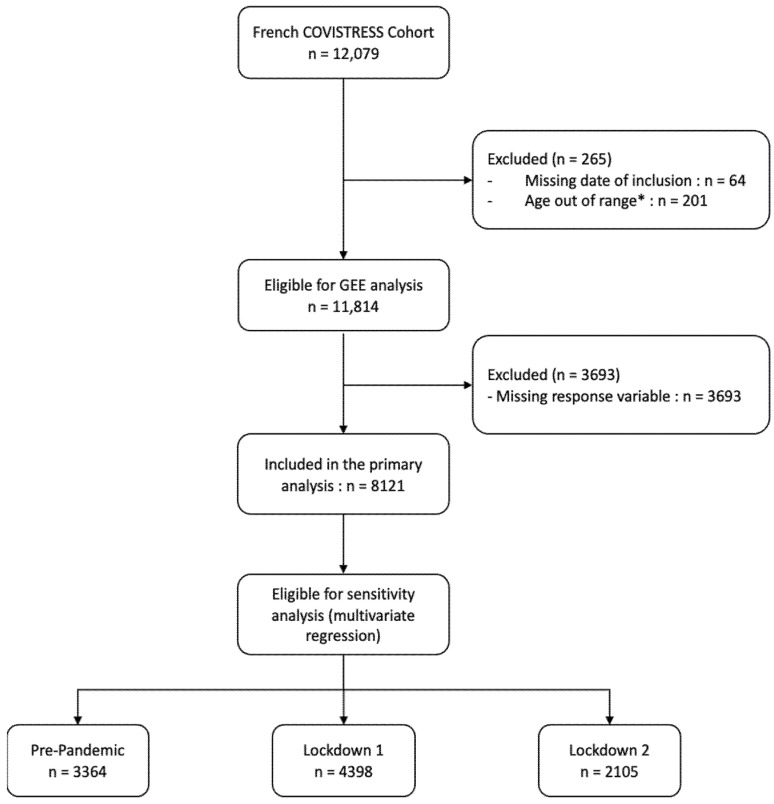
Flowchart of participants. Abbreviations: *: participants aged < 18 and > 75 were excluded from the analysis.

**Figure 2 ijerph-19-09113-f002:**
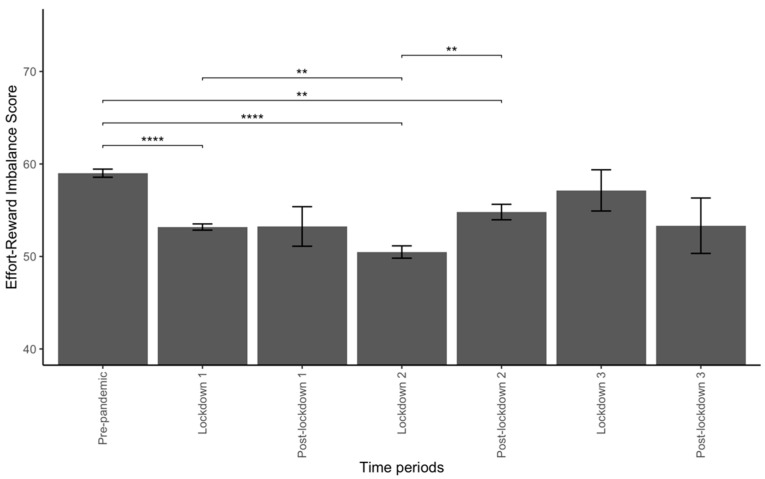
Effort-Reward Imbalance (ERI) score across the periods of interest. Abbreviations: **: <0.01; ****: <0.0001.

**Figure 3 ijerph-19-09113-f003:**
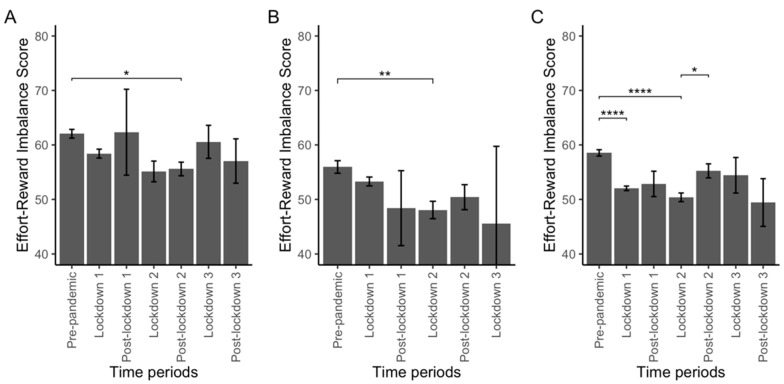
Effort-Reward Imbalance (ERI) score across the periods of interest according to occupations. Plot (**A**): in medical professionals, Plot (**B**): in paramedical professionals, Plot (**C**): in other workers. Abbreviations: *: <0.05; **: <0.01; ****: <0.0001.

**Figure 4 ijerph-19-09113-f004:**
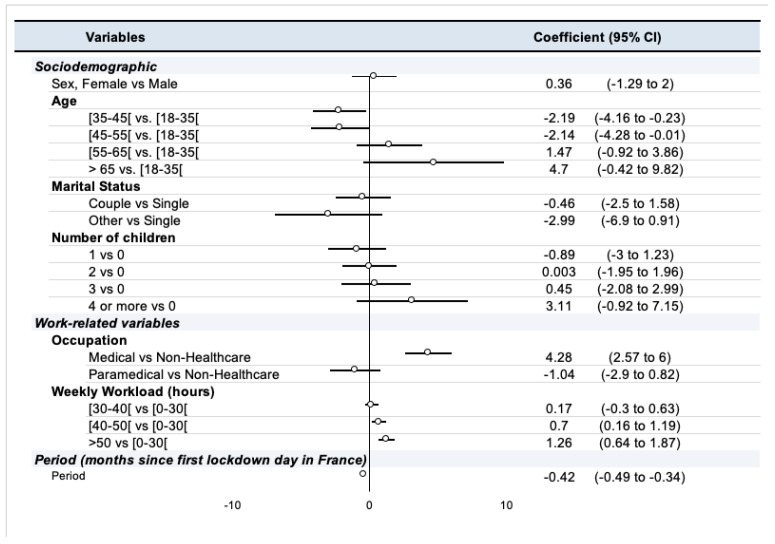
Summary of GEE model with exchangeable covariance structure (GEE-ex).

**Table 1 ijerph-19-09113-t001:** Participants’ characteristics, in the whole cohort and by periods of interest. *: The Pre-pandemic column aggregates the records in which the ERI at the pre-pandemic period was available. The weekly workload during the pandemic periods is thus absent from this pre-pandemic column.

		Time Period							
	Overall	Pre-Pandemic *	Lockdown 1	Post-Lockdown 1	Lockdown 2	Post-Lockdown 2	Lockdown 3	Post-Lockdown 3	*p*-Value
	(*n* = 8121)	(*n* = 3364)	(*n* = 4398)	(*n* = 209)	(*n* = 2105)	(*n* = 1186)	(*n* = 138)	(*n* = 85)	
**Gender**									
Male	1957 (24.1%)	859 (25.5%)	1002 (22.8%)	59 (28.2%)	450 (21.4%)	381 (32.1%)	30 (21.7%)	35 (41.2%)	<0.001
Female	6143 (75.6%)	2496 (74.2%)	3385 (77.0%)	149 (71.3%)	1650 (78.4%)	802 (67.6%)	108 (78.3%)	49 (57.6%)	
Missing	21 (0.3%)	9 (0.3%)	11 (0.3%)	1 (0.5%)	5 (0.2%)	3 (0.3%)	0 (0%)	1 (1.2%)	
**Age**									
under 35	2358 (29.0%)	773 (23.0%)	1483 (33.7%)	67 (32.1%)	449 (21.3%)	269 (22.7%)	57 (41.3%)	33 (38.8%)	<0.001
35–45	2243 (27.6%)	893 (26.5%)	1247 (28.4%)	52 (24.9%)	571 (27.1%)	325 (27.4%)	28 (20.3%)	20 (23.5%)	
45–55	2050 (25.2%)	940 (27.9%)	1018 (23.1%)	56 (26.8%)	623 (29.6%)	313 (26.4%)	27 (19.6%)	13 (15.3%)	
55-65	1257 (15.5%)	640 (19.0%)	562 (12.8%)	29 (13.9%)	404 (19.2%)	226 (19.1%)	23 (16.7%)	13 (15.3%)	
above 65	213 (2.6%)	118 (3.5%)	88 (2.0%)	5 (2.4%)	58 (2.8%)	53 (4.5%)	3 (2.2%)	6 (7.1%)	
**Marital Status**									
as_single	812 (10.0%)	752 (22.4%)	27 (0.6%)	1 (0.5%)	509 (24.2%)	228 (19.2%)	27 (19.6%)	20 (23.5%)	<0.001
as_couple	5045 (62.1%)	2384 (70.9%)	2442 (55.5%)	114 (54.5%)	1436 (68.2%)	895 (75.5%)	98 (71.0%)	60 (70.6%)	
other	310 (3.8%)	114 (3.4%)	182 (4.1%)	11 (5.3%)	84 (4.0%)	28 (2.4%)	4 (2.9%)	1 (1.2%)	
Missing	1954 (24.1%)	114 (3.4%)	1747 (39.7%)	83 (39.7%)	76 (3.6%)	35 (3.0%)	9 (6.5%)	4 (4.7%)	
**Number of Children**									
0	2656 (32.7%)	949 (28.2%)	1589 (36.1%)	79 (37.8%)	578 (27.5%)	335 (28.2%)	50 (36.2%)	25 (29.4%)	<0.001
1	1298 (16.0%)	523 (15.5%)	720 (16.4%)	24 (11.5%)	356 (16.9%)	168 (14.2%)	19 (13.8%)	11 (12.9%)	
2	2456 (30.2%)	1100 (32.7%)	1252 (28.5%)	60 (28.7%)	722 (34.3%)	372 (31.4%)	30 (21.7%)	20 (23.5%)	
3	975 (12.0%)	473 (14.1%)	459 (10.4%)	24 (11.5%)	267 (12.7%)	190 (16.0%)	18 (13.0%)	17 (20.0%)	
4 or more	306 (3.8%)	164 (4.9%)	131 (3.0%)	3 (1.4%)	81 (3.8%)	76 (6.4%)	9 (6.5%)	6 (7.1%)	
Missing	430 (5.3%)	155 (4.6%)	247 (5.6%)	19 (9.1%)	101 (4.8%)	45 (3.8%)	12 (8.7%)	6 (7.1%)	
**Occupation**									
other	5580 (68.7%)	2060 (61.2%)	3266 (74.3%)	163 (78.0%)	1514 (71.9%)	537 (45.3%)	59 (42.8%)	41 (48.2%)	<0.001
medical	1196 (14.7%)	814 (24.2%)	328 (7.5%)	21 (10.0%)	225 (10.7%)	506 (42.7%)	72 (52.2%)	44 (51.8%)	
paramedical	1345 (16.6%)	490 (14.6%)	804 (18.3%)	25 (12.0%)	366 (17.4%)	143 (12.1%)	7 (5.1%)	0 (0%)	
**Working hours pre-pandemic (per week)**									
<30	351 (4.3%)	333 (9.9%)	0 (0%)	0 (0%)	249 (11.8%)	83 (7.0%)	10 (7.2%)	9 (10.6%)	NA
>50	506 (6.2%)	490 (14.6%)	0 (0%)	0 (0%)	158 (7.5%)	304 (25.6%)	25 (18.1%)	19 (22.4%)	
30–40	1517 (18.7%)	1461 (43.4%)	0 (0%)	0 (0%)	1061 (50.4%)	399 (33.6%)	39 (28.3%)	18 (21.2%)	
40–50	659 (8.1%)	641 (19.1%)	0 (0%)	0 (0%)	316 (15.0%)	273 (23.0%)	43 (31.2%)	27 (31.8%)	
Missing	5088 (62.7%)	439 (13.1%)	4398 (100%)	209 (100%)	321 (15.2%)	127 (10.7%)	21 (15.2%)	12 (14.1%)	
**Weekly Working Hours during the Pandemic Period**									
<30	947 (11.7%)	-	507 (11.5%)	27 (12.9%)	276 (13.1%)	109 (9.2%)	17 (12.3%)	11 (12.9%)	<0.001
>50	813 (10.0%)	-	203 (4.6%)	13 (6.2%)	199 (9.5%)	341 (28.8%)	35 (25.4%)	22 (25.9%)	
30–40	4027 (49.6%)	-	2538 (57.7%)	106 (50.7%)	974 (46.3%)	353 (29.8%)	39 (28.3%)	17 (20.0%)	
40–50	1407 (17.3%)	-	677 (15.4%)	39 (18.7%)	345 (16.4%)	289 (24.4%)	34 (24.6%)	23 (27.1%)	
Missing	927 (11.4%)	-	473 (10.8%)	24 (11.5%)	311 (14.8%)	94 (7.9%)	13 (9.4%)	12 (14.1%)	

## Data Availability

The data presented in this study are available on request from the corresponding author.

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
