# Peer review of "The Evolution of Effort-Reward Imbalance in Workers during the COVID-19 Pandemic in France—An Observational Study in More than 8000 Workers"

_ijerph, 2022, doi:10.3390/ijerph19159113_

Round 1

Reviewer 1 Report

Well-described paper. Clear methodological and results section. There are some minor comments for clarity and soundness.  

1. Introduction section could be a bit elaborated. Though there may not be enough published literature on this topic, published information from the WHO, CDC could be referred to elaborate on the content.
2. It would be good to know what other occupations are included in the category of 'others'. In the statistical section or while choosing the model fit, it would be good the check the correlation with some of the occupations listed in the 'others' if they are correlated even by chance.
3. Methods and results sections are clearly described.
4. Most of the association seems to be negatively associated beside a few of the variables. Do you have some strong reasons for the association between the risk associated with e.g. children in the highest category (though the CI seems to be too wide)
5. Association with occupations was observed. It would be good to know briefly about the associations with some interesting variables such as no.of children and age (>65 years).
Discussion sections could be improved (the current version still includes the most interesting information)

Reviewer 2 Report

This paper presents a cross-sectional study focusing the perception of a large population of French workers regarding work-related stress, expressed through Effort-Reward Imbalance, during the Covid-19 pandemic.

This is a relevant and up-to-date research topic, and in my opinion this paper introduces relevant information with adequate scientific soundness. The results obtained are consistent with the objectives of the study. The methodology is clearly described and, to the best of my knowledge, is adequate for the research’s objectives.

I have the following comments / improvement suggestions:

1-    Please improve consistency regarding the designation of ERI: is it Effort/Reward Imbalance or Effort-Reward Imbalance? I believe it should be this second option, since this variable does not represent a ratio, as suggested by the first option.

2-    The numbers in Table 1 and Figure 1 are not consistent (e.g. Table 1: N=11874; Figure 1: eligible = 11814)

3-    Table 1: why no reference to data regarding the pre-pandemic status?

4-    Still regarding the data on Figure 1 and Table 1: In figure 1 you say that the participants regarding the time period ‘lockdown 1’were 6439, but only 4398 were eligible for analysis. Therefore, in Table 1, the data presented should refer only to these 4398 eligible participants, and not to the entire sample of 6439. The same procedure should be adopted for the other time periods, only eligible answers considered for analysis should be presented in Table 1.

5-    In Figures 2 and 3, what is the meaning of the *, **, and ****?

Reviewer 3 Report

A very interesting paper to review. Thank you for submitting.

Abstract:

N = 11,000+ but the actual number of included cases for analysis was less, please correct;

Uses term "Doctors" vs. "paramedical professionals." these terms in the text are used inconsistently. please revise and use consistent terms.

Introduction:

This section contained fewer references than the discussion referred to. Please review this inconsistency and correct.

Methods:

Terms were not clearly defined; these included sex, martial status and types of workers, medical, paramedical and other professions.

The term sex was not inclusively used. Did you mean assigned sex at birth or were you referring to only people identifying as male or female cis gendered and not transgendered, lesbian, gay, bisexual, queer? Who was in the other category?

Marital Status was confusing. There seemed to be single, couple and then other, who were in these categories?

Professional categories was confusing. In the abstract you referred to Doctors and paramedical professionals and other professions. But in the methods you referred to only medical and paramedical professions and then other professions. Who were in these categories. Later in the discussion you refer to Doctors and Nurses. Please clarify throughout.

Referring to data as categorical or quantitative. Quantitative data includes the variables that can be measured. Categorical includes ordinal and nominal, and can be assigned with numbers. Continuous data can be subdivided multiple times and still carry meaning, like minutes, temperature, etc. Please clarify the type of variables correctly. I think you are using quantitative data that are categorical and continuous. Your statistical analysis matches to these types of variables.

Statistical Analysis was adequately done for the variable of interest.

Discussion:

the ERI score from before the pandemic is 'recall" biased and not memorization biased. Memorization bias refers pre- and post- written or oral or survey type tests for an intervention that are not spaced very far apart and the participants can memorize their answers and use them again on the next test. Please change to the correct term.

There are multiple assumptions about Nurses (I think paramedical professionals) about burnout, turnover intentions, moral trauma and deleterious patient outcomes. The references in the discussion as well as this information are new additions not included in the introduction. Either refer to them in the introduction or eliminate them from the discussion. I suggest elimination from the discussion as you did not measure any of the creative topics that you were inferring.

You make an hypothesis from you data that was not really based in the data and was a leap to state. Since, you did not clarify that your study would be hypothesis generating in the introduction or methods, then this should be removed from the discussion.

You used the term "extrapolation" to the general population. The correct statement would be, "this study has relatively good 'generalization' to Doctors and Nurses and other professionals in France's health care system." Please consider changing.
